# Economic Feasibility of Chemical Weed Control in Soybean Production in Serbia

**Maja Meseldžija** [1,*] **, Miloš Rajković** [2] **, Milica Dudić** [1] **, Milica Vranešević** [1] **, Atila Bezdan** [1]**, Aleksandar Jurišić** [1] **and Branka Ljevnaić-Mašić** [1]

1 Faculty of Agriculture, University of Novi Sad, Dositej Obradović Square 8, 21000 Novi Sad, Serbia; milica.dudic@polj.uns.ac.rs (M.D.); milicaraj@polj.uns.ac.rs (M.V.); bezdan@polj.uns.ac.rs (A.B.); aca@polj.uns.ac.rs (A.J.); brana@polj.uns.ac.rs (B.L.-M.)

2 Institute of Field and Vegetable Crops, Maksima Gorkog 30, 21000 Novi Sad, Serbia; milos.rajkovic@nsseme.com

* Correspondence: maja@polj.uns.ac.rs; Tel.: +381-21-4853-448

**Abstract:** The aim of this study was to investigate the efficacy and phytotoxicity of herbicides in combinations and to determine their economic feasibility in the soybean crop. The trials were placed at two locations, Pobeda and Budisava (Serbia), during 2016 and 2018, organized in a randomized block design with four replicates of all herbicide combinations (metribuzin + S-metolachlor; imazamox + oxasulfuron + typhenesulfuron-methyl; metribuzin + S-metolachlor + imazamox + oxasulfuron + typhenesulfuron-methyl; and bentazon + imazamox + typhenesulfuron-methyl). At the first location, 16 weed species were determined, while in the second location, 14 were determined. The highest reduction in the number of weeds was achieved at the first location, using a combination of herbicides metribuzin + S-metolachlor + imazamox + oxasulfuron + typhenesulfuron-methyl with a total efficacy that ranged from 96.98% to 97.40%. Only on the second location bentazone + imazamox + tifensulfuron-methyl showed passable phytotoxicity on soybean, during both years. Based on the established economic justification, in the combinations of metribuzin + S-metolachlor + imazamox + oxasulfuron + typhenesulfuron-methyl, yield was increased for 2350 kg/ha or 1.91 times more, according to the untreated variant at first location. The economic injury level at the most effective combination of herbicides was 52.70 weeds/m$^2$.

**Keywords:** soybean; herbicides efficacy; weeds; economic thresholds; crop yield

## 1. Introduction

In 2050, the human population is projected to increase by 30% to about 9.2 billion [1]. With this increased population density, demands for food production are expected to increase within 70% in developing countries, in regard to meat, milk products, and grains for livestock feed [1,2]. In many regions, crop productivity may be increased by improved water and soil management, high-yielding varieties, fertilization, and other cultivation techniques. However, increased yield potential of crops is often associated with a greater risk of pest attack, leading to an increase in absolute losses and loss rates [3]. Major challenges to agricultural production are to reduce current yield losses caused by weeds, pests, and pathogens [1]. Weeds are considered to be the most harmful to agricultural production; besides affecting agrobiodiversity, they indirectly affect the crop production, by competing with the crops for resources, protecting crop pests, reducing yield and quality, and consequently increasing processing costs [4]. Therefore, weed management is essential for agricultural production and could play an important role in achieving future food production goals [5].

Soybean (*Glycine max* (L.) Merr.) is a wonder crop of the twentieth-century and an important source of protein and vegetable oil for human and animal consumption. Today, the largest soybean producers of the world are the USA, Brazil, and Argentina [6,7]. The soybean contains approximately 40%–45% protein and 18%–22% oil, and it is a rich source of vitamins and minerals, with a favourable composition of the amino acids. For this reason, soybean is recommended for use in human consumption as a part of cholesterol-free diets [8]. In addition to this, soybean can improve soil fertility by fixing atmospheric nitrogen from the air [9], and its oil is being used for biodiesel [10]. In the semiarid and subhumid regions of the world, water stress is a major factor limiting soybean production [11]. In our country, climate has a great impact on plant growth and development, so light, heat, air, and water often represent limiting factors [12]. Economic feasibility and selection of most effective measures in weed management, during the crop season, can be the key to maximizing yields. Serbia has favorable conditions for growing soybeans though proper cultivation technology, and selection of variety is a prerequisite for economically viable soybean production. In 2016, average soybean yields in Serbia were at the level of world yields average, while production amounted to 576,446.00 t [13].

Heavy weed infestations in soybeans highly interfere with the timeliness and efficiency of harvests, and if not controlled, cause severe losses in crops [14]. In soybean production, almost 37% of attainable production is endangered by weed competition compared to pathogens, viruses, and pests [2]. The grain yield reduction due to the weed infestation in soybean varies from 27% to 84%, depending on type of soil, seasons, and intensity of weed infestation in India [15]. Pereira et al. [16] state that weed infestation that occurred in their research caused 90.42% of losses in grain yield in soybean. Weeds not only reduce soybean yields through their competition for nutrients, light, and moisture, but they can also severely reduce harvest efficiency [17]. In Europe, the most common weed species found in soybean are *Amaranthus retroflexus* (L.), *Chenopodium album* (L.), *Solanum nigrum* (L.), *Polygonum persicaria* (L.), *Cirsium arvense* (L.) Scop., *Echinochloa crus-galli* (L.) P. Beauv., and *Sorghum halepense* (L.) Pers. [18]. In addition to these weeds, Serbia has *Ambrosia artemisiifolia* (L.), *Xanthium strumarium* (L.), *Abutilon theophrasti* Medik., *Datura stramonium* (L.), and *Hibiscus trionum* (L.) [19]. Like all other major crops, the prolonged presence of weeds in soybean can result in significant yields and quality losses, so early weed control is necessary to achieve economically acceptable yields [20]. According to Hrustic et al., losses of soybean yield in delayed elimination of weeds is reduced by 25–30% [21].Good soybean weed control involves utilizing all available methods (cultural, mechanical, physical, biological, and chemical) and combining them in an integrated weed management (IWM) system. Today, producers are increasingly turning to herbicide treatments due to labour shortages, high manpower costs, and large areas. Thus, chemical weed control is necessary to decrease cost and to increase productivity, so the advantages of herbicide use are high efficiency, and the presence of selective products soybean at the lowest cost, compared to other available weed control methods [15]. In Serbia, the pre-emergence (pre.em.) herbicides, such as metribuzin, terbuthylazine, flumioxazine, clomazone, pendimethalin, S-metolachlor, and dimethenamide, as well as post-emergence (post.em.) herbicides, such as clethodim, cycloxydim, fenoxaprop-p-ethyl, fluazifop-p-butyl, quizalofop-p-ethyl, propaquizafop, imazamox, oxasulfuron, and tifensulfuron-methyl, have been registered for weed control in the soybean [22]. Weeds can be most effectively managed in soybeans with a well-planned program that includes a detailed analysis of the field situation [23]. The aim of this study was to examine the efficacy of different herbicide combinations in soybean and their phytotoxicity on a crop, and also to assess the economic feasibility of chemical weed control.

## 2. Materials and Methods

Field trials were conducted on the experimental fields at Pobeda (45°45′0″ N; 19°49′0.12″ E; altitude 97 m) and Budisava (45°16′31.8″ N; 19°59′22.8″ E; altitude 86 m), in Serbia, during 2016 and 2018, in the soybean crop. Soybean variety 'Balkan' a I maturity group from Serbia was sown in the first decade of April. This variety is adapted for cultivation in our climate and records the stability and adaptability of yield [12]. A herbicides' efficiency trail was set up by a random block system,

with elementary plots of 25 m$^2$ in 4 repetitions, according to the standard method EPPO/OEPP [24,25]. Control plot, not treated with herbicides, was also included in both localities. Pre.em. herbicides were applied 2 days after sowing (April 13), and post.em. herbicides were sprayed on May 6 and 20, during 2016. In 2018, soybean was sown on April 21, and herbicides combinations were applied directly after sowing (April 21), and post.em. herbicides were sprayed on May 18 and 30. Tests with herbicides (Table 1) were made depending on the presence and dominant weed species. The weed species were identified according to Tutin et al., Josifović and Jávorka and Csapody [26–28]. Herbicide treatments at Pobeda included application of two pre.em. herbicides in combination: metribuzin (Sencor; Bayer AG, Leverkusen, Germany) and S-metolachlor (Dual Gold 960; Syngenta, Basel, Switzerland), applied after sowing and before germination of soybean. Experiment included application of three post.em. herbicides: imazamox (Pulsar 40; BASF Agro, Wadenswil, Switzerland), oxasulfuron (Dynox; Agromarket, Serbia), and typhenesulfuron-methyl (Harmony 75 WG; DuPont International Operations, Grand Saconnex, Switzerland), with the addition of a wetting agent (0.1% Dupont Trend 90). This herbicide combination was applied at the second trifoliate leaf stage of soybean, in split application, which means half the applied dosis in the first and the other half amount after 10 days of the first treatment. In Budisava, treatments included application of three post.em. herbicides: bentazone (Galbenon; Galenika-Fitofarmacija, Belgrade, Serbia), imazamox (Pulsar 40; BASF Agro, Wadenswil, Switzerland), and typhenesulfuron-methyl (Harmony 75 WG; DuPont International Operations, Grand Saconnex, Switzerland), applied in combination at the first trifoliate leaf stage of soybean, also with the addition of a wetting agent (Inex). All applied combinations in the research are biologically compatible, and they have wide use in soybean production. The main data on the tested herbicides and their amounts are presented in Table 1. Herbicides were applied by using a back sprayer 'Solo' with consumption of 300 L of water per hectare. The effectiveness of herbicide treatments was rated 15 (first assessment) and 30 (second assessment) days after the treatment, and, at the same time, assessment of phytotoxicity was made. Weeds were identified and counted using a 1 m$^2$ frame randomly placed in the treated and untreated plots. Based on the obtained data, herbicide effectiveness was calculated by the formula Dodel loc. cit. Janjić [29] and represents a relative ratio between the number of destroyed weeds compared to the number of weeds in the control. Phytotoxic effects of herbicides on crop plants were evaluated by using the European Weed Research Council (EWRC) scale from 1 to 9 (Table 2).

**Table 1.** Tested herbicides.

| First Location—Pobeda | | | | |
|---|---|---|---|---|
| Variants | Preparations | Active Ingredient | Applieddoses | Time of Application |
| 1 | Control | - | - | - |
| 2 | Sencor WG 70 + Dual Gold 960 EC | metribuzin + S-metolachlor | 0.3 l/ha + 1.2 l/ha | pre.em. |
| 3 | Pulsar 40 + Dynox + Harmony 75WG*2 | imazamox + oxasulfuron + thifensulfuron-methyl | 0.5 l/ha + 40 g/ha + 5 g/ha*2 | post.em. |
| 4 | Sencor WG 70 + Dual Gold 960 EC + Pulsar 40 + Dynox + Harmony 75WG | metribuzin + S-metolachlor + imazamox + oxasulfuron + thifensulfuron-methyl | 0.3 l/ha + 1.2 l/ha + 0.5 l/ha + 40 g/ha + 5 g/ha*2 | pre.em. post.em. |
| Second Location—Budisava | | | | |
| 1 | Control | - | - | - |
| 2 | Galbenon + Pulsar 40 + Harmony 75WG + Inex | bentazone + imazamox + thifensulfuron-methyl | 2 l/ha+1 l/ha+8 g/ha | post.em. |

**Table 2.** European Weed Research Council (EWRC) rating scale for phytotoxicity.

| EWRC Score | Crop Tolerance | Damaged Plants (%) |
|:---:|:---:|:---:|
| 1 | No effect | 0 |
| 2 | Very slight effects; some stunting and yellowing just Visible | 1 |
| 3 | Slight effects; stunting and yellowing; effects reversible | 2 |
| 4 | Substantial chlorosis and or stunting; most effects probably reversible | 5 |
| 5 | Strong chlorosis/stunting; thinning of stand | 10 |
| 6 | Increasing severity of damage | 25 |
| 7 | Increasing severity of damage | 50 |
| 8 | Increasing severity of damage | 75 |
| 9 | Total loss of plants and yield | 100 |

An economic threshold (*ET*) for managing weeds in crops may be defined as the weed population at which the cost of control is equal to the value of crop yield attributable to that control [30]. *ET* is taken to prevent an increasing weed population from reaching economically damaging levels, which is the economic injury level (*EIL*).

The *ET* (Equation (1)) and *EIL* (Equation (2)) were calculated by using the following equations [31]:

$$ET(weed/m^2) = \frac{C}{(V \times D)} \tag{1}$$

where *C* is control costs, including application (€/ha), *V* is value of crop (€/t), and *D* is weed damage (t/ha for each weed/m$^2$).

$$EIL(weed/m^2) = \frac{C \times N}{(V \times I)} \tag{2}$$

where *N* is number of weeds per unit area, *V* is market value per production unit, and *I* is the percent yield loss.

Control costs (CC) was calculated, using the following equation (Equation (3)):

$$EIL(weed/m^2) = \frac{C \times N}{(V \times I)} \tag{3}$$

where NA is the number of herbicide applications, AC is application costs (€/ha per application), and IC is herbicide costs per application (€/ha).

Weed competitiveness was quantified with the concept of competitive index (CI), a relative scale of weed competitiveness. Values of CI were calculated for broadleaf weed species based on weed emergence, using Equation (4):

$$EIL(weed/m^2) = \frac{C \times N}{(V \times I)} \tag{4}$$

where *A* is the measured variable for observed weed species (weed emergence plants/m$^2$), B is the measured variable of the most competitive weed (with highest number of weeds/m$^2$), and K is a constant with a value of 10 [32]. Weed emergence was determinate at 15 and 30 days after treatments. Weeds were rated on a scale of 1–10, with a score of 10 for the most competitive weed. This index is used to calculate competitive load (CL) for each weed in the crop. CL (Equation (5)) is defined as the product of the number of weeds of a single species (n$_1$) per unit area (1 m$^2$) multiplied by that species CI value [30].

$$EIL(weed/m^2) = \frac{C \times N}{(V \times I)} \tag{5}$$

Then the total competitive load (TCL) is calculated by summing the CL values for the individual species, using Equation (6):

$$EIL(weed/m^2) = \frac{C \times N}{(V \times I)} \tag{6}$$

The TCL values were used to estimate percent yield loss from a multispecies weed population. The difference in herbicides efficacy were determined by ANOVA and LSD-test, using STATISTICA 13.2 software (Dell™ Statistica™ 13.2 University License, Novi Sad, Serbia).

## 3. Results and Discussion

The differences in the composition of weed vegetation were determined based on a two-year survey conducted at two locations. Identification and determination of weeds are made in order to establish the herbicides efficacy on certain weed species. At the first location (Pobeda), 16 weed species were identified and determined. The experimental field was infested with various weed species, including annual, perennial broadleaf, and grass weed species. The dominant broadleaf weeds were *Abutilon theophrasti* Medik, *Amaranthus retroflexus* L., *Ambrosia artemisiifolia* L., *Chenopodium album* L., *Chenopodium hybridum* L., *Convolvulus arvensis* L., *Datura stramonium* L., *Galeopsis tetrahit* L., *Galium aparine* L., *Hibiscus trionum* L., *Polygonum convolvulus* L., *Solanum nigrum* L., *Sinapis arvensis* L., and *Xanthium strumarium* L., while the most frequent grass weeds were *Echinochloa crus-galli* (L.) Beauv., and *Sorghum halepense* (L.) Pers. A smaller number of grass weeds was observed in the years of investigation. The number of weeds and the efficacy of applied herbicides during the all efficacy assessment in soybean are shown in Table 3.

**Table 3.** Efficacy of used herbicides in soybean on the first location.

| Weed Species | First Assessment | | | | | | |
|---|---|---|---|---|---|---|---|
| | 1 | 2 | | 3 | | 4 | |
| | No/m$^2$ | No/m$^2$ | Ce (%) | No/m$^2$ | Ce (%) | No/m$^2$ | Ce (%) |
| *Abutilon theophrasti* Medik. | 0.75 | 0.25 | 66.66 | 0 | 100 | 0 | 100 |
| *Amaranthus retroflexus* L. | 4.50 | 1.75 | 61.11 | 0 | 100 | 0 | 100 |
| *Ambrosia artemisiifolia* L. | 1.0 | 0 | 100 | 0 | 100 | 0 | 100 |
| *Chenopodium album* L. | 1.5 | 1.0 | 33.33 | 0 | 100 | 0 | 100 |
| *Chenopodium hybridum* L. | 0.75 | 0.25 | 66.66 | 0 | 100 | 0 | 100 |
| *Convolvulus arvensis* L. | 0.50 | 0.25 | 50.00 | 0.25 | 50.00 | 0.25 | 50.00 |
| *Datura stramonium* L. | 2.75 | 0.75 | 72.72 | 0.25 | 90.90 | 0 | 100 |
| *Echinochloa crus galli* (L.) Beauv. | 4.50 | 0.50 | 88.88 | 0.25 | 94.44 | 0 | 100 |
| *Galeopsis tetrahit* L. | 0.50 | 0 | 100 | 0 | 100 | 0 | 100 |
| *Galium aparine* L. | 0.50 | 0 | 100 | 0 | 100 | 0 | 100 |
| *Hibiscus trionum* L. | 2.75 | 0.50 | 81.81 | 0.25 | 90.90 | 0.25 | 90.90 |
| *Polygonum convolvulus* L. | 0.50 | 0.25 | 50.00 | 0 | 100 | 0 | 100 |
| *Solanum nigrum* L. | 1.25 | 0.50 | 60.00 | 0 | 100 | 0 | 100 |
| *Sinapis arvensis* L. | 0.75 | 0.25 | 66.66 | 0 | 100 | 0 | 100 |
| *Sorghum halepense* (L) Pers. rhizome | 3.25 | 0.25 | 92.30 | 0 | 100 | 0 | 100 |
| *Sorghum halepense* (L) Pers.-seed | 12.50 | 0.75 | 94.00 | 0 | 100 | 0.25 | 98.00 |
| *Xanthium strumarium* L. | 0.75 | 0.50 | 33.33 | 0.25 | 66.66 | 0.25 | 66.66 |
| Total number of weeds | 38.50 | 7.75 | | 1.25 | | 1.00 | |
| Total efficacy | - | 79.87% | | 96.75% | | 97.40% | |
| Phytotoxicity | - | 1 | | 1 | | 1 | |

**Table 3.** *Cont.*

| Weed Species | Second Assessment | | | | | | |
|---|---|---|---|---|---|---|---|
| | **1** | **2** | | **3** | | **4** | |
| | No/m$^2$ | No/m$^2$ | Ce (%) | No/m$^2$ | Ce (%) | No/m$^2$ | Ce (%) |
| *Abutilon theophrasti* Medik. | 1.00 | 0.25 | 75.00 | 0 | 100 | 0 | 100 |
| *Amaranthus retroflexus* L. | 4.75 | 0.75 | 84.21 | 0 | 100 | 0 | 100 |
| *Ambrosia artemisiifolia* L. | 1.25 | 0.25 | 80.00 | 0 | 100 | 0 | 100 |
| *Chenopodium album* L. | 1.25 | 0.75 | 40.00 | 0 | 100 | 0 | 100 |
| *Chenopodium hybridum* L. | 0.50 | 0 | 100 | 0 | 100 | 0 | 100 |
| *Convolvulus arvensis* L. | 0.75 | 0.25 | 66.66 | 0.25 | 66.66 | 0.25 | 66.66 |
| *Datura stramonium* L. | 3.50 | 0.50 | 85.71 | 0 | 100 | 0 | 100 |
| *Echinochloa crus galli* (L.) Beauv. | 4.25 | 0.50 | 88.23 | 0.25 | 88.23 | 0 | 100 |
| *Galeopsis tetrahit* L. | 0.25 | 0 | 100 | 0 | 100 | 0 | 100 |
| *Galium aparine* L. | 0.50 | 0 | 100 | 0 | 100 | 0 | 100 |
| *Hibiscus trionum* L. | 3.25 | 0.25 | 92.30 | 0 | 100 | 0.25 | 92.30 |
| *Polygonum convolvulus* L. | 0.50 | 0.25 | 50.00 | 0.25 | 50.00 | 0 | 100 |
| *Solanum nigrum* L. | 0.75 | 0.25 | 66.66 | 0 | 100 | 0 | 100 |
| *Sinapis arvensis* L. | 0.25 | 0 | 100 | 0 | 100 | 0 | 100 |
| *Sorghum halepense* (L) Pers.-rhizome | 3.00 | 0.25 | 91.66 | 0.25 | 91.66 | 0 | 100 |
| *Sorghum halepense* (L) Pers.-seed | 13.75 | 0.75 | 94.54 | 0.50 | 96.36 | 0.50 | 96.36 |
| *Xanthium strumarium* L. | 1.00 | 0.50 | 50.00 | 0.25 | 75.00 | 0.25 | 75.00 |
| Total number of weeds | 41.50 | 5.75 | | 1.75 | | 1.25 | |
| Total efficacy | - | 86.14% | | 95.78% | | 96.98% | |
| Phytotoxicity | - | 1 | | 1 | | 1 | |

1 Control. 2 metribuzin + S-metolachlor. 3 imazamox + oxasulfuron + thifensulfuron-methyl. 4 metribuzin + S-metolachlor+imazamox + oxasulfuron + thifensulfuron-methyl. Ce (%) efficacy coefficient.

*S. halepense* (L.) Pers. is shown in Table 3, separate from rhizomes and from the seeds, to distinguish the effectiveness of the herbicide on the individual. The investigated combinations of herbicides significantly reduced the number of present weeds at Pobeda. After the first assessment, the tested post.em. herbicide combinations of metribuzin at 0.3 l/ha + S-metolachlor at 1.2 l/ha showed good efficacy on *A. artemisiifolia* (100%), *G. tetrahit* (100%), *G. aparine* (100%), and *S. halepense* (of rhizome and seed 92.30 and 94.00%). Satisfactory efficacy of these two herbicides after the first assessment was on *H. trionum* (81.81%) and *E. crus-galli* (88.88%), while after, the second was on *A. theophrasti* (75%), *A. retroflexus* (84.21%), *A. artemisiifolia* (80%), *D. stramonium* (85.71%), and *E. crus-galli* (88.23%). Low efficacy of this herbicides after the first assessment was on *A. theophrasti* (66.66%), *A. retroflexus* (61.11%), *Ch. album* (33.33%), *Ch. hybridum* (66.66%), *C. arvensis* (50.00%), *D. stramonium* (72.72%), *P. convolvulus* (50.00%), *S. nigrum* (60.00%), *S. arvensis* (66.66%), and *X. strumarium* (33.33%), and after the second assessment on *Ch. album* (40.00%), *C. arvensis* (66.66%), *P. convolvulus* (50.00%), *S. nigrum* (66.66%), and *X. strumarium* (50.00%). Belfry et al. [33] reported that S-metolachlor + metribuzin controlled more than 94% of *Setaria viridis* (L.) P. Beauv., and *Ch. album*, as well as *A. artemisiifolia* at two weeks after soybean emergence, while according to Duff et al. [34], S-metolachlor + metribuzin controlled 59% and 91% of *Amaranthus rudis* Sauer eight weeks after treatment.

Split application of imazamox + oxasulfuron + thifensulfuron-methyl shows good efficacy for both broadleaf and grass weeds, except on *C. arvensis* L. and *X. strumarium* L. During the first assessment, the highest efficacy in weed control was achieved by a combination of metribuzin + S-metolachlor + imazamox + oxasulfuron + thifensulfuron-methyl. These herbicides were highly effective on all abundant weeds, except for the species *C. arvensis* and *X. strumarium*. During the second assessment, the efficacy of all tested herbicides was slightly lower in comparison to the first assessment, which is given that the residual effect of herbicide decreases over time. The residuals half-life (DT$_{50}$) of tested herbicides are different values (imazamox (167 days), oxasulfuron (less than

14 days), thifensulfuron-methyl (20–157 days), metribuzin (29–48 days), and S-metolachlor (7–37 days)) [22].

Total efficacy of the tested pre.em. herbicides metribuzin 0.3 l/ha + S-metolachlor 1.2 l/ha during the first and the second assessment was a little lower, and ranged between 79.87% and 86.14%, while total efficacy for herbicides combination imazamox + oxasulfuron + typhenesulfuron-methyl during the first and the second assessment was between 96.75% and 96.98%. The best efficiency (97.40%–96.98%) in weed control after the first and the second assessment was achieved by using the combination of herbicides metribuzin + S-metolachlor + imazamox + oxasulfuron + typhenesulfuron-methyl in split application. Usually, one herbicide application in soybean production is not enough to control all weed species and does not provide long-term protection, although efficient [35]. These are strong reasons for the application of the herbicide combinations in weed management, with special attention to their costs. Nelson et al. [36] stated that tank mixtures of imazamox with thifensulfuron reduced dry weight of *A. artemisiifolia* more than thifensulfuron alone. During 2016, the tested combination of herbicides did not have a phytotoxic effect on soybean (Table 3).

On the second location (Budisava), a total of 14 weed species were identified and determined. The most numerous weed species were *S. nigrum* L., *R. caesius* L., *Ch. Album* L., *H. trionum* L., and *S.halepense* (L.) Pers. The number of weed species and the efficacy of applied herbicides at the second location in soybean are shown in Table 4.

**Table 4.** Efficacy of used herbicides in soybean on the second location.

| Weed Species | First Assessment | | | Second Assessment | | |
|---|---|---|---|---|---|---|
| | 1 | 2 | | 1 | 2 | |
| | No/m$^2$ | No/m$^2$ | Ce (%) | No/m$^2$ | No/m$^2$ | Ce (%) |
| *Ambrosia artemisiifolia* L. | 2 | 0 | 100 | 3.25 | 0.25 | 92.31 |
| *Solanum nigrum* L. | 4 | 0.25 | 93.75 | 2.75 | 0 | 100 |
| *Amaranthus retroflexus* L. | 1.5 | 0 | 100 | 1.25 | 0 | 100 |
| *Chenopodium album* L. | 3.25 | 0 | 100 | 4.00 | 0.25 | 93.75 |
| *Convolvulus arvensis* L. | 2.25 | 0.5 | 77.77 | 1.75 | 0.5 | 71.43 |
| *Hibiscus trionum* L. | 3.25 | 0 | 100 | 3.50 | 0 | 100 |
| *Xanthium strumarium* L. | 0.75 | 0 | 100 | 1.00 | 0.25 | 75 |
| *Rubus caesius* L. | 3.5 | 0.5 | 85.71 | 2.75 | 0.5 | 81.82 |
| *Datura stramonium* L. | 1 | 0 | 100 | 1.75 | 0.25 | 85.71 |
| *Abutilon theophrasti* Medik. | 0,75 | 0 | 100 | 0.50 | 0 | 100 |
| *Sinapis arvensis* L. | 0.25 | 0 | 100 | 0.50 | 0 | 100 |
| *Cirsium arvense* (L.) Scop. | 2.5 | 0 | 100 | 2.25 | 0.50 | 77.78 |
| *Sorghum halepense* (L.) Pers. | 3 | 0.25 | 91.66 | 4.25 | 0.5 | 88.24 |
| *Setaria glauca* (L.) P.B. | 2.25 | 0 | 100 | 1.75 | 0 | 100 |
| Total number of weeds | 30.25 | 1.5 | | 31.25 | 2.75 | |
| Total efficacy | - | 95.04% | | - | 90.43% | |
| Phytotoxicity | - | 2 | | - | 2 | |

1 Control. 2 bentazone + imazamox + thifensulfuron-methyl. Ce (%) efficacy coefficient.

In Budisava, the highest efficacy in weed control was achieved by a combination of herbicides bentazone + imazamox + thifensulfuron-methyl. Tested herbicides in a combination had good efficacy on all abundant weeds, except of the perennial broadleaf weeds *C. arvensis* and *R. caesius*, which efficacy was satisfactory. Total efficacy of these herbicides was good and amounted to 95.04%. After the second assessment, herbicides combination had good efficacy on six broadleaf weeds and one grass weed, while satisfactory efficacy was on *C. arvensis*, *X. strumarium*, *R. caesius*, *D. stramonium*, *C. arvense*, and *S. halepense*, with a total efficiency of 90.43%. After herbicides treatment, the number of weeds per unit area were significantly reduced during the study period when compared with control.

During the study years, combinations of herbicides metribuzin + S-metolachlor + imazamox + oxasulfuron + typhenesulfuron-methyl did not have a phytotoxic effect, while on a tested plot treated

with imazamox + tifensulfuron-methyl in the amount of 1 l/ha + 8 g/ha, passable phytotoxicity was determined. The phytotoxicity symptoms were manifested in the form of transient chlorosis of the leaves but had no significant effect on the development of the soybean, because the crop in the tested combinations (metribuzin + S-metolachlor + imazamox + oxasulfuron + typhenesulfuron-methyl) had the highest yield. According to Nadasy et al. [37], imazethapyr had no harmful effects on soybean varieties on either rate, while metolachlor applied at the double dose caused significant damage, including a reduction in leaf area, fresh leaf weight and phytotoxicity symptoms. Although soybean usually recovers from herbicide injuries as the season progresses, producers question if crop stress from post.em. herbicide injury, in the end, results in reduced soybean yield [38]. The post.em. application of tested herbicides was effective for control of weeds in soybean, more so than the application of metribuzin + S-metolachlor. Application of pre.em. herbicides, separately, could not complete protection of soybeans during the longer period from three to four weeks, so there is a need for use of foliar herbicides (alone or in combinations) [39]. Significant differences between the control, the untreated variant, and treated variants in terms of herbicides efficacy were observed (Tables 5 and 6), which indicates the efficacy of the applied combination of herbicides, due to the fact that a larger number of weeds per m$^2$ was observed on the control plot than on the treated surfaces.

**Table 5.** Statistical analysis of the data after the first and the second assessment at Pobeda.

| | | First Assessment | | | | | | Second Assessment | | | | | |
| No. | Variant | I | II | III | IV | $\overline{X}$ | Sd | I | II | III | IV | $\overline{X}$ | Sd |
|---|---|---|---|---|---|---|---|---|---|---|---|---|---|
| 1 | Control | 40 | 36 | 34 | 44 | 38.50 | 3.84 | 49 | 39 | 37 | 41 | 41.50 | 4.5 |
| 2 | Sencor EG 70 + Dual Gold 960 EC | 1 | 2 | 0 | 1 | 1.00 | 0.70 | 5 | 8 | 7 | 3 | 5.75 | 1.92 |
| 3 | Pulsar 40 + Dynox + Harmony 75WG*2 | 11 | 7 | 8 | 5 | 7.75 | 2.16 | 1 | 4 | 1 | 1 | 1.75 | 1.29 |
| 4 | Sencor WG 70 + Dual Gold 960 ECPulsar 40 + Dynox + Harmony 75WG*2 | 2 | 1 | 2 | 0 | 1.25 | 0.83 | 1 | 2 | 1 | 1 | 1.25 | 0.43 |
| | | LSD$_{0.05}$ = 1.45 | | | | | | LSD$_{0.05}$ = 1.24 | | | | | |

I, II, III, and IV = replications. $\overline{X}$ = the mean value of the number of weeds per m$^2$ from four replicates. Sd = standard deviation. LSD = the least significant difference.

**Table 6.** Statistical analysis of the data after the first and the second assessment in Budisava.

| No. | Variant | I | II | III | IV | $\overline{X}$ | Sd |
|---|---|---|---|---|---|---|---|
| 1 | Control | 31 | 34 | 26 | 30 | 30.25 | 2.86 |
| 2 | Pulsar 40 (1l/ha) + Galbenon (2l/ha) + Harmony (8g/ha) | 2 | 1 | 1 | 2 | 1.50 | 0.50 |
| | | LSD$_{0.05}$ = 4.10 | | | | | |

I, II, III, and IV = replications. $\overline{X}$ = the mean value of the number of weeds per m$^2$ from four replicates. Sd = standard deviation. LSD = the least significant difference

According to the obtained data during 2016 and 2018, EIL, CC and CI values were calculated for dominant broadleaf weeds (*A. teofrasti*, *A. retroflexus*, *A. artemisiifolia*, *Ch. album*, *Ch. hybridum*, *C. arvensis*, *D. stramonium*, *G. tetrahit*, *G.aparine*, *H. trionum*, *P. convolvulus*, *R. caesius*, and *S. nigrum*, *S. arvensis*, *X. strumarium*). Economic analysis of chemical weed management methods was carried out on the expenditure under different herbicide treatments. The calculated values for CI and CL are shown in Tables 7 and 8. As a result of that, on control untreated plots at the first location, soybean yield was 1230 kg/ha with 16% of moisture and 14% of impurity, while on treated plots, soybean yield was in the first treatment 1380 kg/ha, in the second 2560 kg/ha, and at the third treatment 3580 kg/ha, with 10% moisture and 3% impurity.

Data averaged over two years of the study were used to estimate the economic viability of each tested herbicide combinations, in the aim to estimate the profitability of weed control. Application costs were calculated based on fuel cost per ha, labor cost, and depreciation of machines, while IC presented as cost of herbicides per application.

Control costs were CC1 = 23.36 EUR/ha, CC2 = 19.93 EUR/ha, and CC3 = 43.29 EUR/ha. Using the formula, the results obtained for the EIL are as follows:

EIL1 = (23.36 EUR/ha × 5.75 weeds/m$^2$) ÷ (0.34 EUR/ha × 13.86%) = 28.50 weeds/m$^2$,

EIL2 = (19.93 EUR/ha × 1.75 weeds/m$^2$) ÷ (0.34 EUR/ha × 4.22%) = 24.30 weeds/m$^2$ and

EIL3 = (43.29 EUR/ha × 1.25 weeds/m$^2$) ÷ (0.34 EUR/ha × 3.02%) = 52.70 weeds/m$^2$.

At the second location, control cost was CC1 = 68.80 EUR/ha, and EIL was

EIL1 = (68.80 EUR/ha × 1.50 weeds/m$^2$) ÷ (0.32 EUR/ha × 4.96%) = 64.91 weeds/m$^2$.

Broadleaf weeds were more competitive than grass weeds, which resulted in much higher CI values for broadleaf species.

**Table 7.** Competitive index and competitive load for broadleaf weeds at Pobeda.

| Weed Species | Average | CI | CL |
|---|---|---|---|
| *Amaranthus retroflexus* | 0.925 | 10 | 9.25 |
| *Datura stramonium* | 0.625 | 6.8 | 4.25 |
| *Hibiscus trionum* | 0.6 | 6.5 | 3.90 |
| *Solanum nigrum* | 0.2 | 4.3 | 0.86 |
| *Chenopodium album* | 0.275 | 3 | 0.83 |
| *Ambrosia artemisiifolia* | 0.225 | 2.4 | 0.54 |
| *Abutilon theophrasti* | 0.175 | 1.89 | 0.33 |
| *Xanthium strumarium* | 0.175 | 1.89 | 0.33 |
| *Chenopodium hybridum* | 0.125 | 1.35 | 0.17 |
| *Convolvulus arvensis* | 0.125 | 1.35 | 0.17 |
| *Galium aparine* | 0.1 | 1.08 | 0.108 |
| *Polygonum convolvulus* | 0.1 | 1.08 | 0.108 |
| *Sinapis arvensis* | 0.1 | 1.08 | 0.108 |
| *Galeopsis tetrahit* | 0.075 | 0.81 | 0.06 |

Weed diversity and abundance of the population have complicated weed management decision models, with priority to incorporate economic thresholds based on the most competitive weed species [40]. The most competitive weed, at the first location, was *A. retroflexus*, whose CL was 10. CI values based on weed emergence for *D. stramonium*, *H. trionum*, *S. nigrum*, *Ch. Album*, and *A. artemisiifolia* were 6.8, 6.5, 4.3, 3.00, and 2.4, for *A. theophrasti*; *X. strumarium* was 1.89, for *Ch. hybridum* and *C. arvensis* 1.35, while for *G.aparine*, *P. convolvulus*, *S. arvensis* was 1.08. The lowest CI was for *G. tetrahit* 0.81 (Table 7). CI values indicate that the most effective combination will be with active ingredients with efficacy on the weeds with the highest values of this parameter (*A. retroflexus, D. stramonium, H. trionum,* and *S. nigrum*). The total competitive load (TCL) was 21.01 weeds/m$^2$. The projected yield loss was obtained by dividing the TCL by 2 to obtain the percentage by which the competing weeds will reduce the yield. The projected yield loss was 10.51%.

**Table 8.** Competitive index and competitive load for broadleaf weeds in Budisava.

| Weed Species | Average | CI | CL |
|---|---|---|---|
| *Chenopodium album* | 0.725 | 10 | 7.25 |
| *Solanum nigrum* | 0.675 | 9.3 | 6.28 |
| *Hibiscus trionum* | 0.675 | 9.3 | 6.28 |
| *Rubus caesius* | 0.625 | 8.6 | 5.37 |
| *Ambrosia artemisiifolia* | 0.525 | 7.2 | 3.78 |
| *Cirsium arvense* | 0.475 | 6.55 | 3.11 |
| *Convolvulus arvensis* | 0.40 | 5.5 | 2.20 |
| *Amaranthus retroflexus* | 0.275 | 3.79 | 1.04 |
| *Datura stramonium* | 0.275 | 3.79 | 1.04 |
| *Xanthium strumarium* | 0.175 | 2.41 | 0.42 |
| *Abutilon theophrasti* | 0.125 | 1.72 | 0.22 |
| *Sinapis arvensis* | 0.075 | 1.03 | 0.08 |

Compared to the first location, at the second, the most competitive weed was *Ch. album* (CI-10). CI for annual weed species such as *S. nigrum*, *A. artemisiifolia*, *H. trionum*, and perennial *R. caesius* was 9.3, 9.3, 8.6, and 7.2. The lowest CI was for *A. theophrasti* and *S. arvensis*, at 1.72 and 1.03. The total CL was 37.07 weeds/m$^2$, while the projected yield loss was 18.54%. The high value of the projected yield loss has an influence on use of triple herbicide combinations with highest CC (bentazone + imazamox + thifensulfuron-methyl), which results in economic feasibility.

Soybean yields have a direct impact on the final financial results in production. Increased yields have a direct impact on the increased area for soybean production, and for the five year period in Serbia, it increased to 26.6% [41]. According to the statistical data, during 2016, the area under soybean were 200,299 ha, with total yields of 2.41 t/ha, while in 2018, the total area was 220,000 ha, with recorded total yields of 3.31 t/ha [42].

In order to increase yields, it is necessary to reduce production costs, and the selection of the most appropriate chemical measures for weed control will be based on a few key factors. Herbicide combinations will be based on the damage caused by weeds in soybean crops, on herbicide selectivities, herbicide efficacy in order to suppress the dominating weed species, and infallible on economic feasibility. Based on the economic feasibility for the application of the tested herbicide combinations and different amounts, at Pobeda, it can be concluded that the highest costs of weed control were in the third treatment (43.29 EUR/ha), where the highest yield of soybean (3580 kg/ha) and the highest efficacy of applied herbicides (96.98%) was obtained, with an economic injury level of 52.70 weeds/m$^2$. At the Budisava location, the tested herbicides combination allowed for smooth growth and development of soybean, with a yield of 3200 kg/ha, while on the untreated plots, yield was 1700 kg/ha, with 13% of moisture and 8% of impurity. At this location, a multi-year average yield for soybean crop was 2520 kg/ha. We can conclude that, in the set trials, the yield was above the multi-year average.

## 4. Conclusions

From the results obtained from the present investigation in soybean crop, 16 weed species were determined at thePobeda, while 14 species were determined in Budisava. The most dominant were annual broadleaf weeds, but beside them, perennial broadleaf, as well as grass weeds, were present. The tested combinations of all preparations had good efficacy on annual broadleaf weeds, and they significantly reduced the number of weeds compared to control. The combination of pre.em. and post.em. herbicides treatment had low efficacy on perennial broadleaf weed *Convolvulus arvensis*. At the first location, the highest efficacy in weed control was achieved by using combinations of herbicides metribuzin + S-metolachlor + imazamox + oxasulfuron+ typhenesulfuron-methyl, with the total efficacy ranging from 96.98% to 97.40%. Total efficacy of the tested herbicides bentazone + imazamox + thifensulfuron-methyl was good 95.04%. Tested combinations of herbicides metribuzin + S-metolachlor + imazamox + oxasulfuron + typhenesulfuron-methyl did not have a phytotoxic effect, while at the second location, herbicides bentazone + imazamox + thifensulfuron-methyl showed passable phytotoxicity on soybean. The results of this study showed linear relations with the highest reduction in the number of weeds, if herbicides were applied in three treatments (pre.em. and split application of post.em. herbicides). The financial performance assessment found that the most effective variant (metribuzin + S-metolachlor + imazamox + oxasulfuron + typhenesulfuron-methyl) in weed control was economically justified, as well. All treatments from the experiment resulted in higher yields of soybean compared to the control. The highest soybean yield was determined by the combination of soil and foliar herbicides (3580 kg/ha), but with the highest costs of weed control (43.29 EUR/ha) in 2016, while in 2018, yield was 3200kg/ha, with control cost of 68.80 EUR/ha.

**Author Contributions:** Conceptualization, M.M., M.R., and M.D.; methodology, M.M. and M.D.; software, B.L.-M.; validation, M.M. and M.D.; formal analysis, M.M., M.R., and M.D.; investigation, M.M., M.D., and A.J.; resources, M.R., M.V., and A.B.; data curation, B.L.-M. and A.J.; writing—original draft preparation, M.M. and M.D.; writing—review and editing, M.R. and B.L.-M.; visualization, M.D. and B.L.-M.; supervision, M.M.; project administration, M.V. and A.B.; funding acquisition, M.R. All authors have read and agreed to the published version of the manuscript.

**Funding:** This work was funded by the WATERatRISK (Improvement of drought and excess water monitoring for supporting water management and mitigation of risks related to extreme weather conditions) project number INTERREG IPA HUSRB/1206/11/0057 Hungary–Serbia.

**Acknowledgments:** We would like to thank Orsolya Ipacs for carefully reading and correcting our manuscript.

**Conflicts of Interest:** The authors declare no conflicts of interest.

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
