# Peer review of "Economic Feasibility of Chemical Weed Control in Soybean Production in Serbia"

_agronomy, doi:10.3390/agronomy10020291_

Round 1

Reviewer 1 Report

No comments

Reviewer 2 Report

no

Reviewer 3 Report

None.

This manuscript is a resubmission of an earlier submission. The following is a list of the peer review reports and author responses from that submission.

Round 1

Reviewer 1 Report

See comments on the attached document.

Author Response

Dear Reviewer,       

Thank you very much for your constructive comments. They are valuable for improving our manuscript. Below we present our answers to your comments.

Point 1: The Reviewer suggested that the title should be specified because weed condition may vary across regions, state, countries, or globe,  line 2-3.

Response 1: Corrected.

Point 2: The Reviewer marked that phytotoxicity was not mentioned in material and methods, line16

Response 2: We mentioned in Material and methods the phytotoxicity evaluation (line 101-103 and scale in Table 2, wich is in line 105) and we also supplemented it (line 102)

Point 3: In line 18, the Reviewer suggested to write ,, repeated what?,,

Response 3: Corrected, the sentence is revised as suggested (line 18).

Point 4: The Reviewer suggested to add the location, line 19-20.

Response 4: Corrected as suggested.

Point 5: The Reviewer suggested ,,these herbicide are not used individually,,  line 22-23

Response 5: Corrected. The sentence is revised as suggested.

Point 6: In line 25, the Reviewer suggested to add the location

Response 6: Corrected as suggested.

Point 7:  In line 26, the Reviewer suggested correction

Response 7: Corrected. The sentence is revised as suggested.

Point 8:  Missing the reference at the end of the sentence, line 30, line 32-33

Response 8: Reference added

Point 9: The Reviewer suggested to revise the section in the introduction, line 31

Response 9: Corrected as suggested

Point 10: The Reviewer suggested that the sentences in line 32 should be connected

Response 10: Corrected as suggested

Point 11: The Reviewer suggested to revise the line 35

Response 11: Corrected as suggested

Point 12: The Reviewer suggested that the line 35-36 should be in the next paragraph

Response 12: Corrected as suggested

Point 13: The Reviewer suggested to revise the line 42-44

Response 13: Corrected as suggested

Point 14: The Reviewer suggested correction, line 47

Response 14: Corrected as suggested.  We added at line 47

Point 15: The Reviewer suggested to add main yield limiting factors, line 47

Response 15: Corrected as suggested.

Point 16: The Reviewer suggested to change the place of line 48-49, and to put it at the end of the paragraph

Response 16: Corrected as suggested. And at that place, we put other sentences

Point 17: The Reviewer suggested to add at the end of sentences in line 51 ,,in India,,

Response 17: Corrected as suggested.

Point 18: The Reviewer noticed the missing reference in line 54

Response 18: Reference is added.

Point 19: The Reviewer suggested in line 57 to add the major weeds in Serbia

Response 19: Corrected as suggested.

Point 20: The Reviewer suggested that the sentences in line 60-63 is irrelevant, while we didn’t combined variety of control methods

Response 20: Corrected as suggested, the sentences are deleted

Point 21: The Reviewer noticed the missing word ,,and,,and comma in the text, line 65

Response 21: Corrected

Point 22: The Reviewer noticed the missing two comma in the text, line 67

Response 22: Corrected

Point 23: The Reviewer noticed in line 72-74, that the sentences are not connected and something is missing, what about crop phytotoxicity and that abbreviations pre.em. and post.em. should explain at first mention

Response 23: Corrected

Point 24 : The Reviewer asked why we used Balkan variety, line 78

Response 24: We thank the Reviewer for the comment. “Balkan” is medium variety, with maturity group I. High yield potential over 5 t/ha. This variety is adapted for cultivation in our climate and records the stability and adaptability of yield. We added in the text.

Point 25: The Reviewer marked the ,,pre and post emergence,, in the text, line 81

Response 25: Corrected, we explaned before in the text (line 86 and 88 in revised version of manuscript) the meanings of abbreviations (Point 23)

Point 26: The Reviewer mentioned that herbicide treatments are confusing in the text, lines 81, 85 and 90

Response 26: Corrected, we explaned the herbicides treatments and split-application, as well

 Point 27: The Reviewer noticed the missing space in text. Line 98

Response 27: Corrected as suggested.

Point 28: The Reviewer saw some missing in text (space), bold first and second line in Table 1.

Response 28: Corrected as suggested.

Point 29: The Reviewer suggested to change the place of the sentences, and to put them at the end. Line 108-109

Response 29: Corrected.

 Point 30: The Reviewer suggested to revise the sentence in line 112.

Response 30: Corrected as suggested.

Point 31: The Reviewer suggested to defined abbreviations the first time they appear in the text, and then using them in the manuscript. Line 114, 117, 120, 124, 130.

Response 31: Corrected as suggested.

Point 32: In line 126, the Reviewer suggested to correct the sentence.

Response 32: Corrected as suggested

Point 33: The Reviewer suggested that in Material and Methods should be explaned the methods for weed species identification and determination, line 137

Response: Corrected as suggested in Material and Methods, and added the reason for which was made

Point 34: The Reviewer suggested changing the place of the sentence, line 138

Response 34: Corrected as suggested

Point 35: The Reviewer suggested an explanation the differences between Sorghum halepense (L.) Pers. from rhizome and seed, Table 3

Response 35: Corrected as suggested, An explanation of individual presentation of rhizome and seed Sorghum halepense (L.) Pers. has been included

Point 36: The Reviewer suggested to revise the sentences, lines 151 and 152

Response 36: Corrected. The sentence is revised as suggested.

Point 37: The Reviewer suggested to revise the sentences, lines 152-156

Response 37: Corrected. The sentence is revised as suggested.

Point 38: The Reviewer noticed that citate in line 156-157 was about herbicides wich we have not used in our study, and suggestion was to delete them.

Response 38: Corrected. We thank the Reviewer for bringing that to our attention.

 Point 39: The Reviewer noticed that “split application” is missing in M&M.

Response 39: Corrected. We already response and split-application, Point 26

Point 40: The Reviewer suggested to remove the part from the sentence, line 162.

Response 40: Corrected as suggested.

Point 41: The Reviewer suggested to add the residuals halflife of tested chemicals, line 165.

Response  41: Corrected as suggested.

Point 42: The Reviewer suggested to corect the sentices, line 171-172

Response 42: Corrected as suggested

Point 43: The Reviewer suggested to change the references and citations, line 172-175

Response 43: Corrected as suggested

Point 44: The Reviewer suggested uses of abbreviations for Latin names of weed species after first mention. Line 178-180

Response 44: Corrected. We thank the Reviewer for bringing that to our attention.

 Point 45: The Reviewer suggested to corect the sentices, line 183-184 and 190

Response 44: Corrected as suggested.

 Point 46: The Reviewer noticed that we didn’t mention the results of phytotoxicity at first location, and  that phytotoxicity was not mentioned in material and methods, line 191-194

Response 46: Corrected as suggested. We already response in Point 2, for the phytotoxicity evaluation (line 101-103 and scale in Table 2, wich is in line 105) and we also supplemented it (line 102)

Point 47: The Reviewer suggested to change the references and citations, line 194-197

Response 47: Corrected as suggested.

Point 48: The Reviewer suggested to add the references, line 198-199

Response 48: Corrected as suggested.

Point 49: The Reviewer suggested to explain which one of weeds were calculated, line 211

Response 49: Corrected as suggested

Point 50: The Reviewer suggested to check the text in conclusion, line 248-249, line 256

Response 50: Corrected as suggested

Point 51: The Reviewer suggested to add herbicide bentazone, because we used a combination of herbicides (bentazone + imazamox + thifensulfuron-methyl), line 259-260

Response 51: Corrected as suggested

Reviewer 2 Report

In general I have no comments for authors. The paper is ok in my opinion.

Author Response

Dear Reviewer,

We appreciate your positive opinion about our manuscript. At the same time, we address all the points made by the other reviewers and revise the manuscript accordingly, made all the corrections which we hope meet with approval.

Reviewer 3 Report

Abstract:

The lowest value should be indicated first. As it is, it is confusing.

Line 21: … with total efficacy ranged from 97.40 to 96.98%

1.Introduction

This sentence is exactly the same that it is included in the scientific reference. The text should be modified. 

Line 41-42: This is also the reason why soybean is recommended for use in human consumption as a part of cholesterol-free diets [6]

Material and methods

According to the text, were two trials conducted? One in Pobeda and the second in Budisava?

Please, add the year in which each trial was done accordingly the location. The description is quite confusing.

Plant protection products were applied in mixture, are the mixed application of herbicides applied in the tests biologically compatible? It is needed to explain it.

The way to calculate competitive load (CL) should be explained. In addition, in the results, the total competitive load (TCL) is commented, however no reference or description was done previously in the text. This issue should be addressed and clarified.

2.1. Data Analysis

The to obtain the value CI should be explained in detailed, mainly the following variables: A and B. The way in which it is explained is confusing and it is not understood how they are obtained.

3.Results and discussion

Brackets should be removed of the line 161.

Main values of effectiveness reached by the herbicide combinations should be indicated in the text for major weeds.

The way to calculate application costs and where the economic data have been obtained should be better explained.

Conclusions:

The lowest value should be indicated first. As it is, it is confusing.

Line 256: …the total efficacy ranged from 97.40 to 96.98%

The conclusions are very brief, the authors must talk more in depth about the efficacy obtained in each of the treatments applied relating the effectiveness with the control cost. Furthermore, in the conclusions nothing is commented about the values obtained from the indices: CI, CL. All results should be commented and related.

Author Response

Dear Reviewer,

Thank you very much for your constructive comments. They are valuable for improving our manuscript. Below we present our answers to your comments.

Point 1: The Reviewer suggested to indicate the lowest value first. Line 21

Response 1: Corrected as suggested.

Point 2: The Reviewer suggested modifying the text in the introduction. Line 41-42.

Response 2: Corrected as suggested.

Point 3: In material and methods, the Reviewer suggested adding the year in which each trial was done according to location, line 76-77

Response 3: Corrected as suggested.

Point  4: The Reviewer suggested adding whether the combinations are compatible in the test.

Response 4: Corrected as suggested, line 120-121 in the revised manuscript

Point 5: The Reviewer noticed that the equations are missing in the material and methods, line 133

Response 5: Corrected. We thank the Reviewer for bringing that to our attention.

Point 6: The Reviewer suggested to obtaine the value CI and variables A and B, line 129

Response 6: Corrected as suggested.

Point 7: The Reviewer suggested to delete the brackets of the line 161.

Response 7: Corrected as suggested.

Point 8: The Reviewer suggested to add the main values of effectiveness reached by the herbicide combinations for major weeds, line 152-156

Response 8: Corrected as suggested.

Point 9: The Reviewer noticed that the economic data and application costs should be better explained, line 216.

Response 9: Corrected as suggested.

Point 10: In line 256, the Reviewer suggested to indicate the lowest value first.

Response 10: Corrected as suggested.

Point 11: The Reviewer suggested that all results should be commented and related in conclusions.

Response 11: Corrected as suggested.
